# Utilizing Genomic Selection for Wheat Population Development and Improvement

**Lance F. Merrick** [1], **Andrew W. Herr** [1], **Karansher S. Sandhu** [1], **Dennis N. Lozada** [2] and **Arron H. Carter** [1,*]

1 Department of Crop and Soil Sciences, Washington State University, Pullman, WA 99164, USA; lance.merrick@wsu.edu (L.F.M.); andrew.herr@wsu.edu (A.W.H.); k.sandhu@wsu.edu (K.S.S.)
2 Department of Plant and Environmental Sciences, New Mexico State University, Las Cruces, NM 88003, USA; dlozada@nmsu.edu
* Correspondence: ahcarter@wsu.edu; Tel.: +1-509-335-6198

**Abstract:** Wheat (*Triticum aestivum* L.) breeding programs can take over a decade to release a new variety. However, new methods of selection, such as genomic selection (GS), must be integrated to decrease the time it takes to release new varieties to meet the demand of a growing population. The implementation of GS into breeding programs is still being explored, with many studies showing its potential to change wheat breeding through achieving higher genetic gain. In this review, we explore the integration of GS for a wheat breeding program by redesigning the traditional breeding pipeline to implement GS. We propose implementing a two-part breeding strategy by differentiating between population improvement and product development. The implementation of GS in the product development pipeline can be integrated into most stages and can predict within and across breeding cycles. Additionally, we explore optimizing the population improvement strategy through GS recurrent selection schemes to reduce crossing cycle time and significantly increase genetic gain. The recurrent selection schemes can be optimized for parental selection, maintenance of genetic variation, and optimal cross-prediction. Overall, we outline the ability to increase the genetic gain of a breeding program by implementing GS and a two-part breeding strategy.

**Keywords:** plant breeding; two-part strategy; recurrent selection; population improvement; product development; optimization; genetic gain; cross-prediction

## 1. Wheat Breeding

Wheat (*Triticum aestivum* L.) is one of the essential cereal grain crops and is the principal cereal grain used for food consumption in the US and most of the world [1]. Global wheat production has increased 342% from 222 million to 760 million tons from 1961 to 2020, and United States wheat production has increased 148% from 34 million to 50 million tons in the same time period [2]. Yet, the world population is projected to reach 9 billion people by 2050, and to account for this growing demand for food, production systems will have to increase productivity by 50% while facing a decrease in agricultural land [3]. Increasing the productivity of food systems partially relies on developing improved cultivars. However, it usually takes 11–12 years to develop new wheat cultivars [4]. Traditionally, after crossing and population development, inbred lines are developed either through self-pollination or doubled haploids. The inbred lines are then phenotyped in headrows and field trials before being selected as parents in the crossing block. This method takes up to four to six years in wheat, depending on the breeding program structure and preference of the breeder. In the Washington State University winter wheat breeding program, for example, the inbred lines are developed through both self-pollination and doubled-haploid production. Headrows are the first stage of phenotyping and occurs in the fourth year, followed by unreplicated preliminary yield trials (PYTs) in the fifth year. In the sixth year, replicated field trials begin at multiple locations and move to state-wide advanced yield trials (AYTs) at additional locations to screen for a variety of traits. Inbred lines can be in replicated yield trials up to

five or more years by the time varieties are ultimately released from the breeding program. Therefore, modern breeding approaches, which can reduce cycle time and increase selection precision, along with the efficient use of genetic variation can be exceedingly important to increase genetic gains [3].

## 2. Genomic Selection

One modern breeding approach, genomic selection (GS), is posed to increase genetic gains and reduce cycle time for complex agronomic traits such as grain yield and disease resistance [5,6]. Meuwissen et al. [7] proposed the idea of simultaneously estimating all markers across the whole genome, regardless of "significance", thereby capturing all markers' effects, and coined this method "genomic selection". The goal of GS is to calculate genomic estimated breeding values (GEBVs). GS is accomplished using a population of individuals with both phenotypic and genotypic data called the training population. The training population is used to create a prediction model to simultaneously estimate allele effects at all loci. A statistical model is created or trained on the training set to estimate model parameters that are used to calculate the GEBVs of individuals with only genotypic data, called the test population [8]. The selection for the advancement in the breeding program is then based on the GEBVs of the lines rather than phenotypic selection (PS), hence the term "genomic selection".

The cost of genotyping has allowed the application of next-generation sequencing (NGS) and revolutionized applied plant breeding. Next-generation sequencing decreased the cost of whole-genome sequencing using markers, such as single nucleotide polymorphisms (SNPs), allowing cost-effective GS [9]. The development of NGS allowed for an exponential advance in genotyping driven by the goal of sequencing different genomes. Sequencing has improved with the implementation of parallel sequencing that allows for polymorphism discovery, gene expression analysis, and population genotyping [10]. One of the most cost-effective genotyping platforms that can be utilized in large populations and is commonly used in breeding programs is genotyping-by-sequencing (GBS). Further, GBS does not need target SNPs to be identified previously, which reduces costs and bias while enabling application across diverse populations [9]. Newer advances in GBS include skim-based GBS, GT–seq, and rAMpSeq [11–13]. Skim-based GBS enables high-resolution genotyping via low-coverage sequencing [11]. Genotype-by-sequencing allows for a low-cost per sample method of producing thousands of molecular markers needed for GS without a reference genome [9]. Genomic selection takes advantage of these dense genome-wide markers created by GBS by simultaneously predicting all markers' effects and avoiding running statistical tests for significance as mentioned previously. Genotype-by-sequencing allows for the genotyping of large populations of potential selection candidates for GS. However, through the use of GS, most of these candidates are discarded and, therefore, low-cost genotyping is vital.

As an approach, GS can improve a breeding program by increasing genetic gain and enhancing trait selection [6,14,15]. Genetic gain, also known as the genetic response ($R$), is calculated by what is known as the breeder's equation, $R = \frac{ir\sigma_A}{t}$, where $i$ is the selection intensity; $\sigma_A$ is the square root of the additive genetic variance; $r$ is the selection accuracy, which is the equivalent to narrow-sense heritability ($h^2$) in PS; $t$ is the cycle time [16,17]. Plant breeders use this equation to increase the genetic gain of their breeding program through subsequent cycles of selection. By increasing one of the components in the numerator (i.e., selection intensity or selection accuracy) or decreasing cycle time, a breeder can increase genetic gain [17]. GS reduces the length of the breeding cycle by selecting based on GEBVs as opposed to phenotypes. Previously, Lorenzana and Bernardo [18] showed a response per cycle for GS that was half that of PS. However, GS can also be implemented in earlier generations, which can be a substitute for phenotyping that is expensive or time intensive. Thus, selections can be made in a shorter time frame and reduced cycle time [14]. In contrast, Rutkoski et al. [6] displayed that PS and GS exhibited similar genetic gain.

However, GS reduced genetic variance, which indicated that maintaining genetic variance needs to be accounted for in a GS breeding pipeline.

Both PS and marker-assisted selection (MAS) are effective for increasing genetic gain for highly heritable traits [8]. Highly heritable traits, such as disease resistance, can be selected early in the breeding program with high accuracy using MAS. Even though MAS has helped improve plant breeding and increased genetic gain, it is less effective when used for polygenic and complex traits. Additionally, the increase in genetic gain using PS is also difficult for complex traits with low heritability [6]. Consequently, selection for important traits with low heritability, such as grain yield, is completed at the later stages of a breeding program. If the environmental effect, such as drought, disease pressure, or other adverse conditions, is high enough, accurate PS will be challenging.

In contrast, GS can theoretically be used to select for any trait at any stage at the breeding program and is up to the breeder's preference, but it can be advantageous when PS and MAS are ineffective such as selecting for grain yield in early generation lines or other low-heritable traits. Further, GS can select for multiple traits at the same time and is advantageous when trying to evaluate and select genotypes based on combinations of yield components, end-use quality, or disease traits [19]. The joint analysis of multiple traits takes advantage of the genetic correlation between the traits, which can increase prediction accuracy, specifically for lowly heritable traits that are genetically correlated with highly heritable traits and, ultimately, increases selection accuracy and genetic variance similar [19–22]. Additionally, GS can increase gain per unit time, increase selection for difficult to measure traits, or assess the performance of individuals in environments where they have not been phenotyped. Finally, GS can reduce the amount of phenotyping that is completed when there are too many lines to phenotype and when seed amounts are insufficient [23].

The implementation of GS into breeding programs is still being explored. Many studies have shown the potential of GS to change traits through higher genetic gain [5,6,24,25]. Wheat breeding programs are still determining the best strategy to implement GS within the pipeline [26]. Reviews have helped guide the integration of GS within wheat breeding programs [3,27,28]. As genotyping costs continue to decrease, GS is starting to be practically cheaper than PS. Nevertheless, various aspects need to be considered first before integrating GS into the breeding pipeline. In order to implement GS and replace PS, breeding programs have to account for selecting on GEBVs instead of phenotypic values, the time it takes for genotyping and DNA extraction, and the use of doubled-haploids (DHs) in rapid-cycle recurrent GS schemes. In addition, previous GS studies focused on selecting lines based exclusively on GEBVs; however, the implementation of phenotypic validation of the top selections still needs to be made, since most prediction accuracies only account for a portion of the variance [26]. Genomic selection allows for the use of GEBVs in lieu of phenotypic data, which allows for the restructuring of the traditional breeding program. Gaynor et al. [29] proposed reorganizing the traditional breeding program into two parts: the product development (PD) component, which is similar to traditional breeding programs; a population improvement (PI) component to utilize recurrent GS. In this review, we explored the integration of GS for a wheat breeding program by redesigning the traditional breeding pipeline to differentiate between PD and PI while optimizing a two-part strategy based on the components of the breeder's equation.

## 3. Product Development

The PD part of wheat breeding programs focuses on developing inbred lines for release as inbred varieties. The PD pipeline is the same as a traditional breeding program and, therefore, is the easiest and most flexible component to change. One of the major changes between the PD pipeline and the traditional breeding program is that lines from PD are not necessarily chosen for population improvement. However, the lines phenotyped in PD will need to be genotyped to update the training population for GS implementation. Stochastic simulations were previously used to compare long-term selection and genetic gain using

various breeding program structures [29]. Gaynor et al. [29] simulated a two-part breeding program that used GS to conduct recurrent selection on $F_1$ plants for PI and additional parental selection in PYTs and a similar program with parental selection in headrows (Figure 1). The two-part breeding programs showed up to 2.47 times the genetic gain than the conventional program with no GS and up to 1.46 times the genetic gain relative to the conventional program with GS alone but no parental selection. Overall, the two-part strategy improved the genetic gain of a breeding program at similar costs and resource allocation.

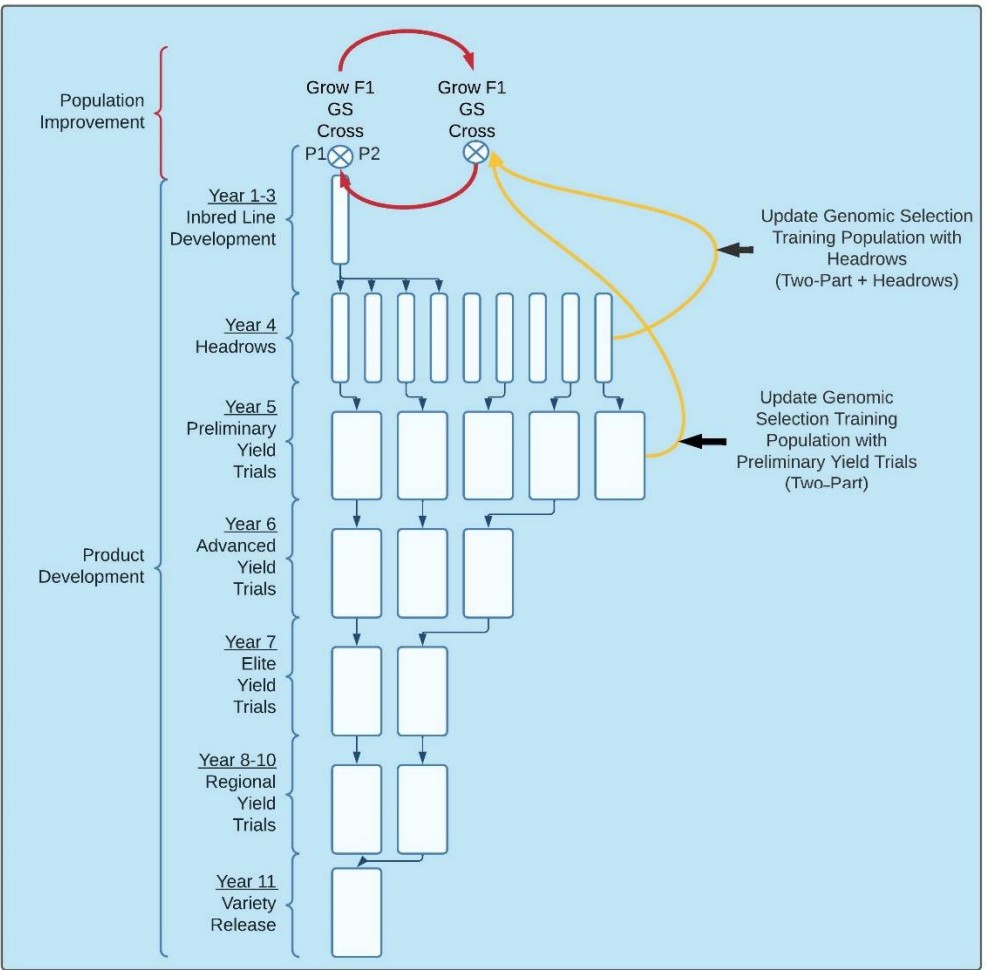

**Figure 1.** Overview of the two-part breeding strategy based on an 11 year breeding program from parental crossing to variety release. The recurrent selection scheme is shown in red arrows using $F_1$s for continuous crossing within-population improvement, whereas the blue lines show the product development (PD) pipeline. The yellow arrows display the possible implementations of GS into the PD pipeline for integration into the recurrent selection scheme, which include utilizing lines from the preliminary yield trials (two-part) or lines from headrows (two-part + headrows) to update the GS training population.

## 3.1. Implementation of GS for Recurrent and Parental Selection

Replacing phenotypes with GEBVs allows for the restructuring of breeding programs. Genomic selection can simply replace phenotypic or MAS for selection purposes [24,30]. However, this strategy does not necessarily increase genetic gain for certain traits, such as grain yield, due to the lack of increase in selection accuracy compared to PS. Therefore, breeders need to determine the best stage to implement GS to increase every aspect of the breeder's equation and maximize genetic gain. GS can be implemented at any stage in order to select lines for the next stage in the breeding program. In general, field trials begin with

many lines with fewer replications and environments; then, the number of lines is sequentially decreased while increasing replications in individual trials and more locations. There are several opportunities to increase the genetic gain by optimizing breeding programs for GS. These include reorganizing field designs, increasing the number of lines evaluated, and leveraging the large amount of genomic and phenotypic data collected across different growing seasons and environments to increase heritability estimates, selection intensity, and selection accuracy [26,31,32]. Optimization for GS implementation enhances genetic gain and can decrease costs.

For parental or recurrent selection in early generations, GS implemented as early as the $F_1$ or $F_2$ generations is effective without including a generation of PS (Figure 2) [26]. Implementing GS as early as possible displayed increases in genetic gain by nearly six times compared to phenotypic selection with no GS implementation [26]. However, implementing GS early increases genotyping costs, whereas using GS in later generations with one stage of PS reduces the number of lines genotyped (Figure 2). The implementation of GS in the $F_3$ and $F_4$ generations displayed a two-fold increase in genetic gain in $F_3$ selection compared to PS in the same stage [26]. Implementation in the $F_3$ displayed a compromise between early implementation with no PS and PS with GS implemented in later stages of the program (Figure 2). Implementing GS in the late stages of the breeding program was shown to have little advantage in increasing genetic gain than PS alone [26]. Additionally, Longin et al. [33] showed through simulation that when genomic selection accuracy is low, one stage of GS should be followed by one stage of PS. Additionally, GS would be recommended only when accuracy is very high. In low-accuracy scenarios, GS would be useful with truncation selection for removing the lowest performing lines.

### 3.2. Implementation of GS for within and across Breeding Cycles

Genomic selection can also be implemented in the PD pipeline across and within breeding cycles for selection and advancement (Figure 3) [15,25,34,35]. Early-generation GS is superior to conventional PS in line breeding and can be strongly improved by including additional information from later generation PYTs and AYTs [15,35]. For instance, phenotypic data from the PYT or AYT can be used to predict advancements across breeding cycles as early as $F_2$, DH, and inbred lines [36,37]. Additionally, the same PYTs and AYTs can be used to predict advancements from the PYT and AYT across breeding cycles or within the same breeding cycle [15,34].

The implementation of GS for selection in PYTs has been extensively studied [25,31,34,35]. The PYT has been intuitively chosen because they are the first time that lines are yield tested and, generally, constitute the largest filtering stage before subjecting lines to resource-demanding replicated multi-location yield trials. Additionally, since the PYT is the first time that lines are generally grown in plots, the seed is limited, resulting in either one or two replications. Therefore, since there is a trade-off between the number of lines being tested and the number of replications, Endelman et al. [31] demonstrated that higher prediction accuracy for GS can be achieved in the PYT by utilizing an unbalanced field design across multiple locations rather than testing all lines in a single location.

As mentioned previously, within-cycle selection can be implemented in most stages of the breeding program. Verges and Van Sanford [35] demonstrated, by utilizing within-cycle selection, that the number of lines in the PYT can be increased by phenotyping half of the lines and using the results to predict the performance of the other half. However, according to Michel et al. [34], predicting across cycles rather than within cycles decreased protein yield and content bias. Additionally, utilizing phenotypes from all breeding cycles increased prediction accuracy compared to within-cycle accuracy [38–40]. Further, Belamkar et al. [25] showed that prediction accuracy could be increased by merging across and within-cycle selection by combining half of the lines in the within-cycle PYT with previous PYT phenotypes. Furthermore, due to the limitations of resources and replications coupled with the moderate prediction accuracy for various traits, GS could be implemented to

altogether remove the phenotyping of PYT and select lines for advancements directly into replicated yield trials or reduce the number of years lines are in the replicated yield trials.

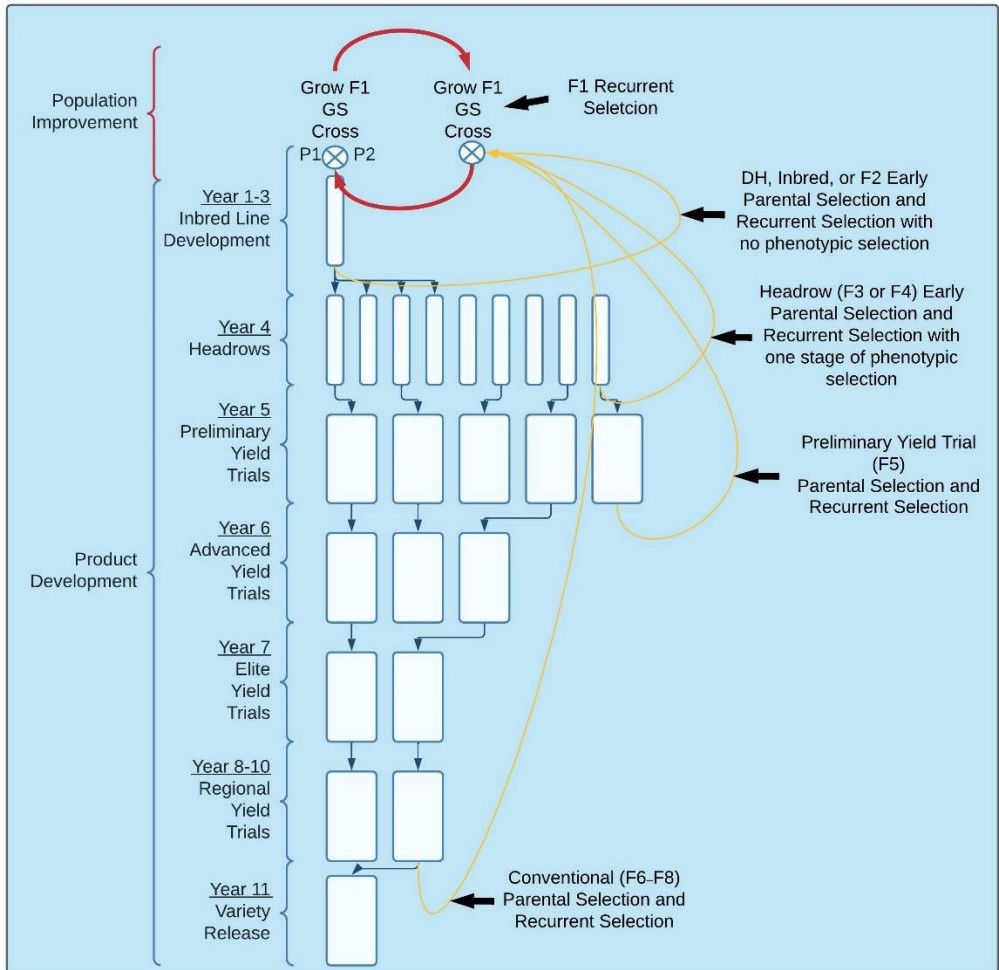

**Figure 2.** The implementation of GS for recurrent and parental selection based on an 11 year breeding program from parental crossing to variety release within the two-part breeding strategy. The recurrent selection scheme is shown in red arrows using $F_1$s for continuous crossing within-population improvement, whereas the blue arrows show the product development (PD) pipeline. The yellow arrows display the possible implementations of GS into the PD pipeline to select lines for integration into the recurrent selection scheme. GS can be implemented as early as the $F_2$ stage, through doubled-haploid production or speed breeding without phenotypic screening. Further, GS can be implemented throughout the PD pipeline with an increasing number of phenotyping stages, from a single stage of phenotyping in headrows to the conventional method of phenotyping up to the 10th year in yield trials.

Regardless of the approach, the implementation of GS can be viewed as a tool to reduce costly phenotyping and allocate resources elsewhere. GS can help aid or replace disease nurseries or select traits that are dependent on the environment for variation [40]. Irrespective of the stage of implementation, design of the breeding program, or trait, simulations can be used to compare the implementation of GS within breeding programs [23]. Throughout this review, many of the studies used simulations including the development of the two-part breeding strategy proposed by Gaynor et al. [29]. There are a number of plant breeding simulation software such as "QU-GENE" [41], "AlphaSim" [42], and "Delta-Gen" [43]. In Gaynor et al. [29], AlphaSim was used due to the fact of its ease, flexibility, and ability to simulate both plant and animal breeding programs. Recently, AlphaSim was

developed into an R package [44]. Plant breeders can now simulate the majority of breeding decisions and GS optimizations before dedicating limited resources to redesigning their breeding programs.

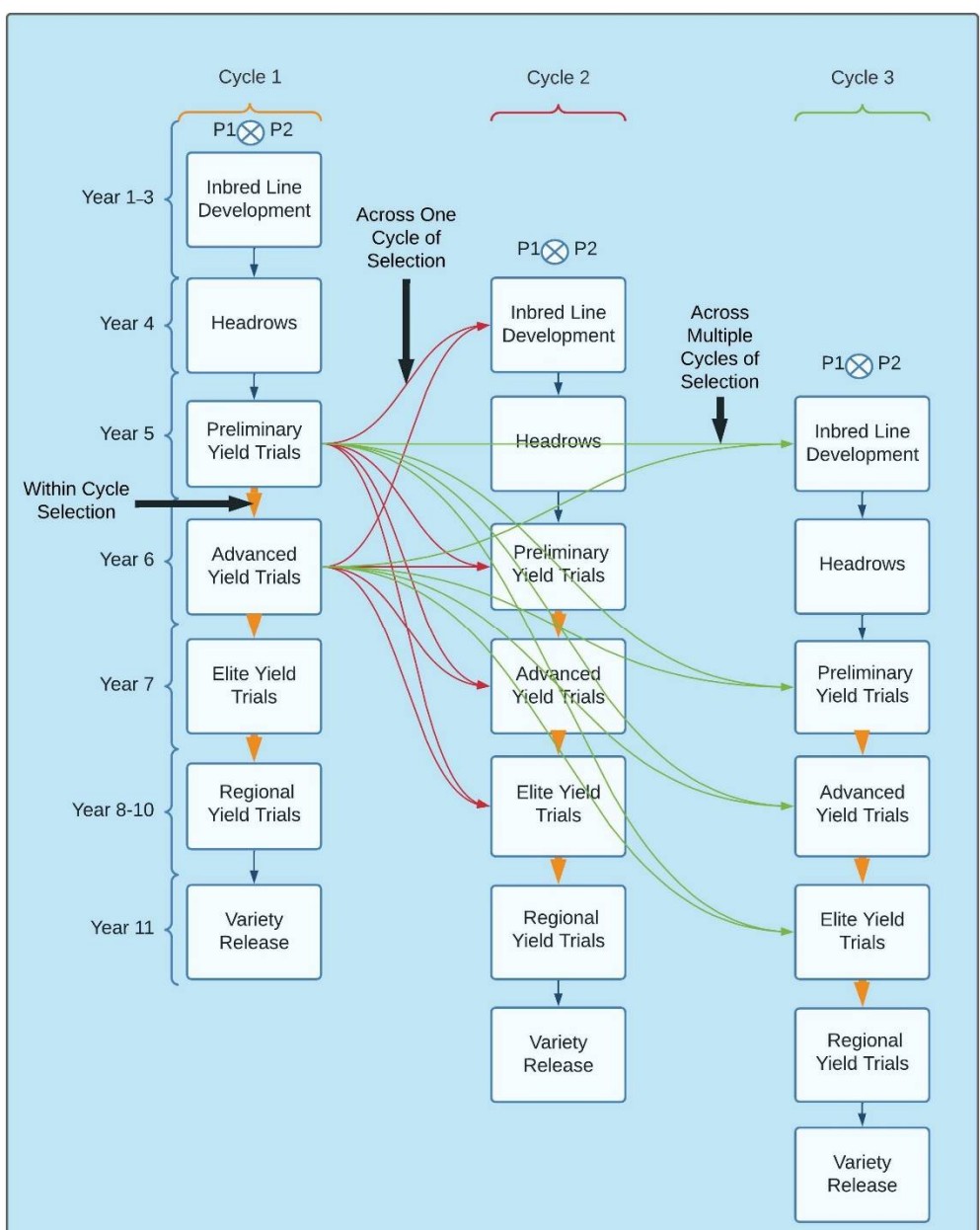

**Figure 3.** The implementation of GS for within and across breeding cycles based on an 11 year breeding program from parental crossing to variety release. The within-cycle selection is shown by the orange vertical arrows within cycles for advancement through various yield trials such as using GS to predict the performance of lines for advancements from the preliminary yield trials to the advanced yield trials compared to the blue arrows for the traditional phenotypic selection. Across cycle selection across a single year is shown using red arrows using preliminary and advanced yield trials to predict inbred lines and yield trials from cycle 1 to cycle 2. Green arrows show the across cycle selection across multiple breeding cycles and are accomplished similarly to selection across a single year but can utilize historical data using multiple cycles for prediction purposes.

## 4. Population Improvement

Population improvement has the greatest opportunity to increase genetic gain by leveraging the major advantage GS has over PS, which is decreasing the cycle time for variety development [14]. The goal of PI is to minimize the selection cycle for identifying parents in the crossing block. GS is adequately poised to break the need for phenotyping new parents by using GEBVs and simply genotyping untested lines without inbreeding or further selection. Therefore, recurrent selection can be accomplished rapidly within a breeding program. As mentioned above, the two-part strategy proposed by Gaynor et al. [29] relies heavily on the recurrent selection scheme for PD.

As discussed previously, Gaynor et al. [29] outlined two strategies for recurrent selection in a two-part breeding program. The two strategies were recurrent selection based on selecting $F_1$ to cross with new parents added to the training population based on inbreds from PYT or based on selection out of headrows. The two-part program was accomplished by randomly dividing the parents' as male and female parents equally. Then, male sterility techniques were applied to conduct open pollination of the females. Seeds were randomly selected from each half-sib family to continue the selection cycle, and the other seeds were used to produce DH lines, with a reduction in DH lines compared to the conventional program. The DH lines were subsequently screened in the PD pipeline, and GS was applied in the PYT for inclusion to the training population, but not the crossing block. The two-part scheme with headrow selection was conducted similarly but with selection out of the headrows instead of PYT. To mitigate increases in genotyping costs, the two-part scheme with headrow selection needs a reduction in the lines per cross. The two-part plus headrow selection outperformed the regular two-part breeding scheme in terms of genetic gain [29].

### 4.1. Selection Scheme

The benefit of the two-part strategy is in large part due to the decrease in cycle time and subsequent increase in genetic gain. By implementing a recurrent selection scheme, the breeding program can achieve the shortest possible cycle time. Most recurrent selection has been based on simulations for theoretical GS strategies in breeding programs (Figure 4).

In one of the first comparisons conducted for GS and marker-assisted recurrent selection (MARS) schemes, two cycles of selection for grain yield in maize (*Zea mays* L.) were simulated [45]. In the initial population development (Cycle 0), two inbred lines were crossed to form an $F_1$ population that was developed into DHs. The DHs showed higher responses than $F_2$ [45]. The best DHs were selected based on testcross performance. The selected DH lines were randomly mated to form $F_1$s for Cycle 1. Genome-wide markers were used in GS, and significantly associated markers were used for MARS. Lines were selected using GS and MARS, and then randomly mated to form Cycle 2. This was repeated to form Cycle 3, where the response to selection was calculated. The simulations displayed that the maximum response was always greater with GS than MARS. Highly heritable traits displayed a 6% increase with GS over MARS; however, for lowly heritable traits, the advantage increased to 18% for GS over MARS. Therefore, for a complex trait with low heritability, GS has a large advantage over MARS. Since selection indices have low heritability due to the combination of multiple traits, GS was further hypothesized to have an advantage over MARS when using selection indices.

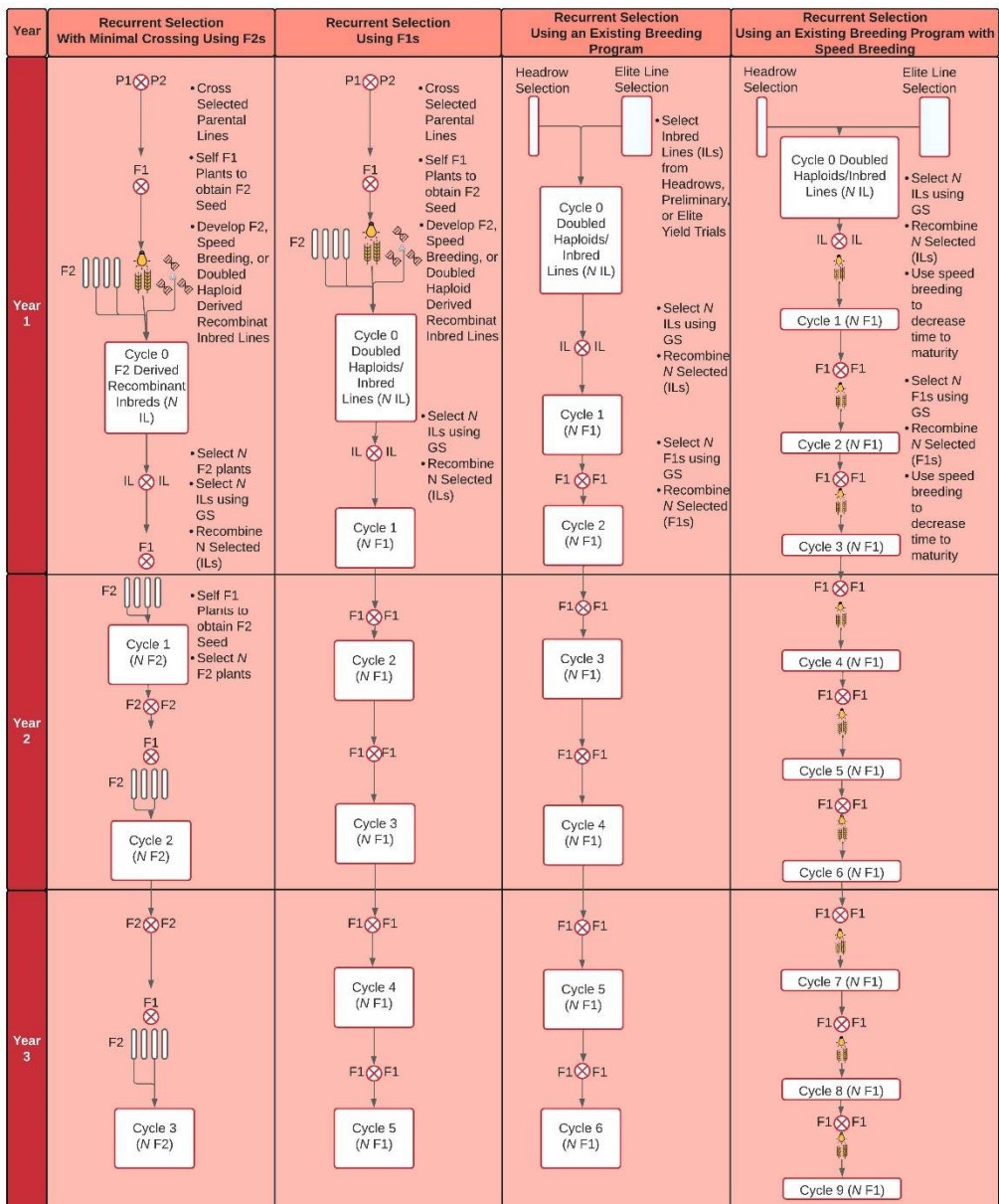

**Figure 4.** Genomic selection recurrent selection schemes comparing the number of recurrent selection cycles achieved over three years with *N* being the number of selected recombinant inbred lines (ILs), doubled-haploids (DHs), $F_1$, or $F_2$ lines. Recurrent selection with minimal crossing using $F_2$ lines consists of developing an initial IL population and then implementing GS to select *N* ILs to recombine, and the $F_1$s are then self-pollinated to form an $F_2$ population that makes up the next recurrent selection cycle. This method can achieve approximately one cycle per year. Recurrent selection using $F_1$s is similar to recurrent selection using $F_2$s. However, the $F_1$s are not self-pollinated, and recurrent selection is implemented on the $F_1$s to make up the next recurrent selection cycle and achieve up to two cycles per year. Further, the number of recurrent selection cycles can be increased by utilizing the existing breeding program and the use of speed breeding to decrease time to maturity for crossing and selection purposes.

Bernardo [46] outlined a GS scheme in inbred crops that minimizes crossing (Figure 4). Using the existing recombinant-inbred lines (RILs) from the breeding program that was evaluated in multi-location year trials, the best RILs were selected based on their phenotypic performance. These selected RILs were crossed, and the $F_1$ progenies were allowed to self-pollinate to $F_2$. The $F_2$ were then grown and genotyped with the best progeny selected

based on GEBVs and crossed once again. These $F_1$s were again self-pollinated to $F_2$s, and the process was repeated. Recurring selection ultimately decreased cycle time and increased genetic gain [46]. This type of crossing has also been called rapid cycling and focuses more on parental selection and crossing rather than later generational cultivar selection. In simulations, the recurrent selection scheme was compared to a recurrent selection scheme without self-pollination (Figure 4). Due to the time it takes to self-pollinate, three cycles of recurrent selection were accomplished in three years using a recurrent selection scheme with self-pollination compared to five cycles in three years using recurrent selection without self-pollination. The response to selection was 81–87% lower than a recurrent selection scheme without self-pollination. However, after three cycles, the response to selection with the recurrent selection with self-pollination improved 4–12% over the response to selection of the recurrent selection without self-pollination. The increase in the response to selection for the self-pollination scheme was due to the increase in the genetic variation from self-pollinating $F_1$ to $F_2$ that can then be exploited by selection. However, the simulations also determined that the number of individuals evaluated in the self-pollination scheme should be twice as large as in the scheme without self-pollination. Further, increasing the selection intensity increased the response to selection. In self-pollinated crops with hand crossing, obtaining large amounts of seed is difficult and, therefore, the selection response is reduced by ~20% compared to selection without self-pollination. However, this can be overcome with an increase in selection intensity and an increase in plants evaluated.

However, recurrent selection can be accomplished in the $F_1$. For example, in intermediate wheatgrass (*Thinopyrum intermedium*), recurrent selection on $F_1$s has been proposed to aid in rapid domestication and improvement. The intermediate wheatgrass program started in 1988 with two cycles of PS for seed size and fertility [47]. It was projected that it would take 20 years to reach the same level of genetic gain of wheat and 100 years to reach the same size [48]. Therefore, a GS scheme with a cycle time of one year was proposed. The proposed scheme starts with $F_1$ seeds from the parent crosses. The $F_1$ lines were then genotyped, and GS was used to determine the best 100 $F_1$, which were then included in the crossing block for the next cycle. A total of 1000–1200 plants were then planted in the field for phenotypic evaluation to update the GS model. The recurrent selection scheme was estimated to increase the rate of genetic gain 2.6 times higher than PS for spike yield and higher for other traits.

The increase in genetic gain between the two-part breeding schemes and the conventional scheme is based on reducing cycle time. In the two-part breeding scheme, two cycles of recurrent selection were accomplished within a single year (Figure 4). The genetic gain can be further improved by increasing the number of cycles per year and further decreasing cycle time. Using rapid cycling via speed breeding, the length of cycle time could be drastically decreased with up to six cycles having been achieved [49]. However, a decrease in the accuracy of GS models and a reduction in genotypic variance can reduce genetic gains while increasing cycles of selection. Further, since GS requires genotypic data, every implementation of GS, especially in recurrent selection and PI, is dependent on the amount of time it takes to genotype the lines. To mitigate this, Hickey et al. [50] recommended that GS could be applied every two or three generations but would need to be supplemented with MAS or PS. Therefore, to maximize the number of recurrent selection cycles implemented with GS, the generation of genotyping, predicting, and crossing needs to be optimized [26].

*4.2. Integration of Germplasm and Maintaining Genetic Variance*

Another limitation of the two-part simulations in Gaynor et al. [29] is the division between the PI and PD in which selected DH lines were only included in the training population but not in the crossing block. In a practical implementation of the two-part strategy, the PI pipeline would become a pre-breeding scheme to rapidly produce lines for the PD pipeline. Further, the inbred lines from the PYT or headrows would be introduced into the crossing block as parents. The two-part system needs to integrate germplasm

and parents from outside the program or via the PD pipeline. Further, within recurrent selection, methods for maintaining genetic variation need to be considered and implemented as discussed in Rutkoski et al. [6]. This is even more important in a practical implementation of the two-part strategy, where enough variation is needed to be maintained for selecting multiple traits throughout the PD pipeline. Genetic variation can be maintained by integrating outside germplasm or through optimal contribution and cross selection. Gaynor et al. [29] suggested that new integrations into the PD pipeline be used as males in order to spread their alleles throughout the population efficiently. Additionally, optimal contribution selection can be integrated to ensure newly introduced alleles are not lost [51,52]. As an approach, optimal contribution selection was developed to balance the contribution and mating of breeding parents for the next cycle to minimize genetic relatedness and inbreeding among parents and progeny [51]. In breeding programs, such as those of CIMMYT, Mexico, the maintenance of genetic diversity is vital for the ability to screen and select lines in developing countries [53]. Optimal cross selection was simulated over a 20 year period and compared against traditional truncation selection with various numbers of cycles per year [54]. The optimal contribution selection utilized AlphaMate to account for optimized selection, genetic diversity, and cross-allocation altogether [54,55]. Using AlphaMate, optimal cross selection increased long-term genetic gain by up to 78% compared to truncation selection on a small number of parents. In addition, a decrease in GS accuracy due to the loss of the genetic relationship between the training and prediction populations was mitigated [54]. The ability to predict the performance of crosses increases the efficiency of genetic and financial resources and, therefore, is a vital part of redesigning a breeding program to focus on GS and population development [53].

*4.3. Optimal Cross-Prediction*

AlphaMate has the flexibility to be used in both cross- and self-pollinated plant species. The software optimizes selection for genetic gain and genetic variation by minimizing group co-ancestry, inbreeding of individual mating, and maximizing group inbreeding. In addition to optimizing selection, breeding programs need to balance short and long-term genetic gain and individual contributions while maintaining genetic diversity [55]. Another software that accounts for the genetic variance for cross-prediction in biparental breeding populations is the "PopVar" package [56] in R. When predicting a cross, it was observed that the mean of the population explained more variation for the mean of the superior progeny than the genetic variance [56]. Likewise, the relevance of the mean was shown in Lado et al. [57] and Yao et al. [58], indicating that the mean of the population was more influential than the genetic variance when predicting crosses for grain yield in wheat. However, the importance of genetic variance was larger in end-use quality traits [57]. Additionally, genetic variance becomes more influential when predicting crosses when the difference in the mean of the parents is low as in the case when crossing elite material. Therefore, it is important to understand the parental materials when predicting and selecting crosses. PopVar predicts the genetic variance by measuring the variance among GEBVs as outlined in Bernardo [59]. In addition, utilizing PopVar or other crossing simulations in recurrent selection and population development allows the imposition of lower selection intensity while maintaining genetic variation to increase genetic gain rapidly.

To maximize the effect of a recurrent selection scheme, methods to select multiple traits efficiently must be developed including, in part, selection index or separate focus on single-trait improvement. For example, Yao et al. [58] showed the use of selection indices using a usefulness rating for grain yield and end-use quality traits composed of weighted values for grain yield, extensibility, and maximum resistance and demonstrated the ability to improve grain yield and end-use quality in wheat simultaneously. Additionally, near-zero or positive correlations were identified in predicted progeny for negatively correlated traits in the parents [58]. PopVar can predict values for correlated traits, which allows for the simultaneous selection of traits. In addition, PopVar predicted correlated responses of multiple traits such as with the negative correlation between grain yield and

deoxynivalenol [56]. In general, when selecting multiple traits, there is a trade-off on the performance of either one trait or both. Nevertheless, PopVar can identify crosses that have a favorable mean for one, with a near-zero mean in the other, or can determine a favorable mean for both as observed previously [58]. Therefore, PopVar can allow for genetic gain for simultaneous selection, even among negatively correlated traits.

In both Yao et al. [58] and Mohammadi et al. [56], common GS models were used to predict marker effects for parents. Cross-predictions utilizing GEBVs are laid out in Endelman [60] using rrBLUP. Additionally, the R package "sommer" [61] can use univariate and multivariate linear mixed models. The "sommer" package has the added capability of integrating dominance and epistatic effects that can be utilized for hybrid selection. For population development, additive effects are more important. Therefore, there is a greater emphasis on utilizing additive effects for sustained genetic gain [56].

Recurrent selection is dependent on identifying parents through the developed inbred lines or the progeny of crosses, such as $F_1$, or after self-pollination using the $F_2$ lines, depending on the recurrent selection scheme. Utilizing GS in parental selection allows for the reduction in cycle time and, therefore, increases genetic gain. Inbred lines can be selected based on traditional self-pollination or DH production described previously. However, inbred lines for parental selection are not needed for the implementation of GS and are an artifact of conventional breeding programs. The selection of $F_1$s or $F_2$s in a recurrent selection scheme have been reported in both empirical and simulation studies [6,45,46]. By selecting non-inbred lines, the cycle time can be drastically reduced, and recurrent selection by means of genotyping, genomic selection, and then crossing can be repeated for rapid cycling, thus accomplishing rapid recurrent selection. A common adage in plant breeding is "plant breeding is a numbers game". This adage relates to the breeder's equation for genetic gain in which increasing the number of lines increases genetic variance and selection intensity. However, breeding programs have limited resources and can only screen so many lines. Therefore, the selection of parents for crossing needs to be efficient by making fewer crosses among the parental lines and effectively changing plant breeding from a "numbers game" to a more precise "chess game". For parental selection, the ability to identify the breeding populations with the highest mean performance prior to making a cross is more important than the number and size of the breeding population [62,63]. Additionally, with better parental selection, the family size of the population can be increased with a decrease in the number of overall populations. Further, Hickey et al. [64] showed an increase in prediction accuracy with the increase in lines within families, whereas Verges and Van Sanford [35] concluded a minimum number of 25 lines per family were needed to stabilize prediction accuracies at the PYT level.

## 5. Real-World Applications

The first recurrent GS experiment in crops was conducted for grain yield and stover quality in maize [5], where GS was compared with MARS (Table 1). In contrast to GS, MARS is based on MAS with significant markers for recurrent selection. A total of three cycles of selection for grain yield and stover quality in maize was compared in the recurrent selection scheme. The scheme started with an initial biparental population (Cycle 0) that was evaluated for traits and genotyped for markers of interest [5]. Multi-trait selection indices were then constructed for grain yield and stover quality. Cycle 0 individuals were ranked according to selection index values, with the top individuals selected and intermated to form Cycle 1. Cycle 1 individuals were genotyped and used to predict the selection index. The Cycle 1 individuals were once again ranked for their predicted values, with the top individuals selected and intermated to form Cycle 2. This was completed one more time to produce Cycle 3. Individuals from Cycles 0–3 were compared and evaluated in field trials. GS displayed a higher stover index and grain yield in Cycle 3 compared to MARS. Additionally, the yield indexes for Cycles 2 and 3 were significantly higher than for Cycle 1 for GS but not for MARS. Further, the majority of gains were accomplished from Cycle 1 due to the Cycle 0 selection being based on PS. Overall, GS resulted in a greater

genetic gain than MARS for selection indices. However, for single traits, the majority of Cycle 2 selections for GS and MARS showed no improvement over Cycle 1. Further, grain yield showed no improvement over Cycle 0.

**Table 1.** Empirical genomic selection (GS) recurrent selection studies.

| Crop | Trait | Cycles | Selection Methods Compared [1] | Gain | Reference |
|------|-------|--------|-------------------------------|------|-----------|
| Maize | Stover Index and Grain Yield | 3 | MARS vs. GS | GS had higher genetic gain compared to MARS | [5] |
| Wheat | Quantitative Adult Plant Stem Rust Resistance | 2 | PS vs. GS | GS had equal rates of genetic gain compared to PS | [6] |
| Wheat | Grain Yield | 1 | GS models | Reproducing kernel Hilbert spaces (RKHS) GS model had the highest realized genetic gain | [53] |
| Barley | Grain Yield and DON | 3 | TP optimization | Optimization algorithms improved accuracy compared to randomly selected TPs | [37] |
| Wheat | Wheat Grain Fructan | 2 | GS with TS vs. GS with (OCS) | OCS and TS had similar genetic gains; OCS retained greater genetic variance | [65] |

[1] GS—genomic selection; MARS—marker-assisted recurrent selection; OCS—optimal contribution selection; PS—phenotypic selection; TP—training population; TS—truncation selection.

Rutkoski et al. [6] reported the first realized GS in wheat and the first using recurrent selection (Table 1), where two cycles of GS to one cycle of PS for quantitative adult plant resistance to stem rust were compared. Accordingly, GS led to equal rates of genetic gain per unit time compared to PS but rapidly decreased genetic variance. The low prediction accuracy in the first cycle of GS led to the lack of increase in genetic gain over PS. CIMMYT also conducted empirical GS in early generations [53], where 200 $F_2$ plants were selected from 40 different crosses (Table 1). The $F_{2:4}$ lines were bulked and tested in a replicated yield trial for two years. The results showed that the RKHS model had the highest accuracy with an increase in grain yield of 7% when comparing the highest GEBV lines to the lowest GEBV lines [53]. Additionally, recurrent selection has been applied in other public breeding programs. For instance, three cycles of selection in a spring six-row barley population were able to increase genetic gains for both grain yield and deoxynivalenol, which are two negatively correlated traits [37]. Further, recurrent selection implemented with optimized contribution selection increased wheat grain fructan content by 34% while controlling the rate of inbreeding [65]. Altogether, previous simulations and actual field validation studies showed recurrent selection using GS to increase genetic gain. Therefore, there can be a significant improvement in gains achieved through selection in both private and public breeding programs through implementation of GS and a two-part breeding program.

## 6. Conclusions

Due to the decreasing costs of genotyping, GS is starting to be practically cheaper than PS, which allows for the use of GEBVs in lieu of phenotypic data. Therefore, the implementation of GS promotes the restructuring of the traditional breeding program. In our review, we explored redesigning a breeding program for GS and the implementation of the two-part breeding scheme. The two-part breeding program was proposed to differentiate

between PI and PD, which showed up to 2.47 times the genetic gain than a traditional breeding program with no GS [54].

The PD pipeline represents the traditional breeding program based on PS for variety release. Since the PD pipeline is similar to the traditional breeding program, it is the easiest and most flexible component to change. The PD not only serves as a pipeline to screen and select lines for variety release, but it is now the source of the training population. Additionally, GS models allow breeders to select lines for implementing into the crossing block from virtually any stage of the breeding program and, therefore, drastically reduce the cycle time for parental selection. Additionally, GS can be implemented to predict lines for advancement within and across breeding cycles and years for the majority of traits. In our review, we discussed the advantages and practicality of utilizing GS in the PYTs. PYTs were intuitively chosen because they represent the first stage that lines are yield tested and constitute the largest stage for screening lines before replicated yield trials. By utilizing the existing structure of the breeding program, the PD pipeline represents the fewest obstacles for the integration of GS and is only limited by the breeder's imagination in how they will redesign and implement GS for screening and selecting lines.

Further, the PI pipeline represents the greatest opportunity to increase genetic gain. By implementing recurrent selection into PI, the genetic gain is effectively limited by the speed and ability to genotype, predict, and cross parental lines and $F_1$s. Recurrent selection is advantageous due to the minimization of cycle time for parental selection. However, due to the fast pace of recurrent selection, careful consideration must be given to maintaining genetic variation and reducing inbreeding. Through the hard work of previous researchers, easy-to-use software, such as AlphaMate and PopVar, have been created to help select parents and predict the performance of crosses for multiple traits simultaneously while utilizing optimal cross-prediction. In contrast to the PD pipeline, the recurrent selection scheme may be vastly different from the traditional breeding program, and that is why the majority of the breeding program will need to be redesigned to maximize the effectiveness of PI and to meet the needs of GS.

Therefore, by utilizing GS and reducing the reliance on PS alone, breeders can redesign their breeding programs and increase genetic gains. This increase in genetic gains will help improve lines and meet the production demands of a growing population. Now it is up to breeders to implement these changes to transition their breeding programs to maximize efficiency and effectively change plant breeding from a "numbers game" to a more precise "chess game" while proving the power of GS approaches.

**Author Contributions:** Conceptualization, L.F.M.; writing—original draft preparation, L.F.M., A.W.H. and K.S.S.; writing—review and editing, L.F.M., A.W.H., K.S.S., D.N.L. and A.H.C.; supervision, A.H.C.; funding acquisition, A.H.C. All authors have read and agreed to the published version of the manuscript.

**Funding:** This research was partially funded by the National Institute of Food and Agriculture (NIFA) of the US Department of Agriculture (Award numbers: 2016-68004-24770 and 2022-68013-36439); Hatch project 1014919; the O.A. Vogel Research Foundation at Washington State University.

**Institutional Review Board Statement:** Not applicable.

**Informed Consent Statement:** Not applicable.

**Data Availability Statement:** Not applicable.

**Conflicts of Interest:** The authors declare no conflict of interest.

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
