# Peer review of "Utilizing Genomic Selection for Wheat Population Development and Improvement"

_agronomy, doi:10.3390/agronomy12020522_

Round 1
Reviewer 1 Report
The review is very innovative, interactive and with very impressive information. I do not have too much to add about it. I only suggest adding some information about global wheat production. See the following link:https://www.fao.org/faostat/en/#data/QCL/visualize
Author Response
- The review is very innovative, interactive and with very impressive information. I do not have too much to add about it. I only suggest adding some information about global wheat production. See the following link:https://www.fao.org/faostat/en/#data/QCL/visualize
Thank you very much for your time and comments. To address your comment, we have added a few statistics using the link you provided for wheat production in the US and Globally on lines 29-32.

Reviewer 2 Report
Minor writing issues are presented. Thus, proofreading would be a good idea
Author Response
- Minor writing issues are presented. Thus, proofreading would be a good idea.
Thank you very much for your time and comments. We have gone through the document and corrected errors. The changes are tracked in the revised document.

Reviewer 3 Report
- The main limitation of the review is to understand how genomic selection (GS) is performed at the molecular level. It would be interesting to show the reader what are the steps to predict the genetic value of selection candidates and, more importantly, at the molecular level how the data are obtained for the predictions. Selection by GS into wheat breeding programs has been carried out with what kind of genetic markers? What are the traits that these markers identify in this crop? Where in the wheat genome are they present? I strongly recommend commenting on these aspects in topic 2 and adding a figure outlining the traits selected by GS in wheat, the main markers used to predict candidates based on the GEBV, and where the markers are present in the wheat genome.
- The conclusions (topic 6) should be focused on the discussion of wheat breeding. This topic is written in a generic way, therefore outside the main scope of the work, which is to explore the integration of GS for a wheat breeding program.
Author Response
- The main limitation of the review is to understand how genomic selection (GS) is performed at the molecular level. It would be interesting to show the reader what are the steps to predict the genetic value of selection candidates and, more importantly, at the molecular level how the data are obtained for the predictions.
Thank you very much for your time and comments. We have added a paragraph in section 2 describing genotyping and markers used in GS from lines 65-85. However, it is beyond the scope of this review to go into great detail about genotyping and sequencing.
- Selection by GS into wheat breeding programs has been carried out with what kind of genetic markers?
Thank you for your comment. We have clarified this concern in the previous comment from lines 66-85.
- What are the traits that these markers identify in this crop?
Thank you for your comment. Discussion of traits based on heritability and effectiveness of GS is on lines 102-124. We have added some clarification on traits, but we clarified GS can theoretically be used to select for any trait at any stage of the breeding program.
- Where in the wheat genome are they present?
Thank you for your comment, but this comment is beyond the scope of this review. GS uses markers from across the entire genome and therefore, the location of individual markers on the genome are not as important. We have added a clarification on line 56 about using markers across the whole genome.
- I strongly recommend commenting on these aspects in topic 2 and adding a figure outlining the traits selected by GS in wheat, the main markers used to predict candidates based on the GEBV, and where the markers are present in the wheat genome.
Thank you for your comment. I believe we have addressed your concerns in the previous comments. We believe that a figure outlining the traits and markers is not necessary since GS can select for any trait. The presence of the markers on the wheat genome is more pertinent to marker-assisted selection than GS since we do not focus on statistical tests for significance of individual markers, and instead use all of the markers available.
- The conclusions (topic 6) should be focused on the discussion of wheat breeding. This topic is written in a generic way, therefore outside the main scope of the work, which is to explore the integration of GS for a wheat breeding program.
Thank you for pointing this out. To address your comment, we have added the main points of the paper into the conclusion to provide a better overview of the integration of GS for a wheat breeding program.
